# A Novel Model of Pathogenesis of *Metarhizium anisopliae* Propagules through the Midguts of *Aedes aegypti* Larvae

**DOI:** 10.3390/insects14040328

**Published:** 2023-03-28

**Authors:** Ricardo de Oliveira Barbosa Bitencourt, Jacenir Reis dos Santos-Mallet, Carl Lowenberger, Adriana Ventura, Patrícia Silva Gôlo, Vânia Rita Elias Pinheiro Bittencourt, Isabele da Costa Angelo

**Affiliations:** 1Graduate Program in Veterinary Sciences, Veterinary Institute, Federal Rural University of Rio de Janeiro, Seropédica 23890-000, RJ, Brazil; 2Oswaldo Cruz Foundation, IOC-FIOCRUZ-RJ, Rio de Janeiro 21040-900, RJ, Brazil; 3Oswaldo Cruz Foundation, IOC-FIOCRUZ-PI, Teresina 64001-350, PI, Brazil; 4Laboratory of Surveillance and Biodiversity in Health, Iguaçu University-UNIG, Nova Iguaçu 28300-000, RJ, Brazil; 5Centre for Cell Biology, Development and Disease, Department of Biological Sciences, Simon Fraser University, Burnaby, BC V5A 1S6, Canada; 6Department of Animal Biology, Institute of Health and Biological Sciences, Federal Rural University of Rio de Janeiro, Seropédica 23890-000, RJ, Brazil; 7Department of Animal Parasitology, Veterinary Institute, Federal Rural University of Rio de Janeiro, Seropédica 23890-000, RJ, Brazil; 8Department of Epidemiology and Public Health, Veterinary Institute, Federal Rural University of Rio de Janeiro, Seropédica 23890-000, RJ, Brazil

**Keywords:** mosquitoes, entomopathogenic fungi, blastospores, conidia

## Abstract

**Simple Summary:**

The mosquito *Aedes aegypti* is the principal vector of multiple arboviruses including dengue (DENV), zika (ZIKV), and chikungunya (CHIKV) that cause major human disease and suffering. Control is based on the application of synthetic insecticides to reduce mosquito populations. However, the overuse of synthetic insecticides has led to mosquito resistance. Studies on the use of entomopathogenic fungi (EPF) such as *Metarhizium anisopliae* to reduce mosquito populations are gaining interest as they are ecologically safe and can kill different mosquito life stages. Despite their lethality towards mosquito larvae, their mechanisms of infection are unclear. In this study, we first evaluated the production of blastospores of three *M. anisopliae* isolates and assessed their effects against *Ae. aegypti* larvae. We assessed, and describe in detail, the mechanism of infection of one fungal isolate against *Ae. aegypti* larvae. Overall, our findings indicate that EPF are able to kill mosquito larvae by infecting the insect midgut, disrupting enterocytes, and causing brush border degradation.

**Abstract:**

We assessed the effect of the entomopathogenic fungus *Metarhizium anisopliae* against *Aedes aegypti*. Conidia of *M. anisopliae* strains CG 489, CG 153, and IBCB 481 were grown in Adamek medium under different conditions to improve blastospore production. Mosquito larvae were exposed to blastospores or conidia of the three fungal strains at 1 × 10^7^ propagules mL^−1^. *M. anisopliae* IBCB 481 and CG 153 reduced larval survival by 100%, whereas CG 489 decreased survival by about 50%. Blastospores of *M. anisopliae* IBCB 481 had better results in lowering larval survival. *M. anisopliae* CG 489 and CG 153 reduced larval survival similarly. For histopathology (HP) and scanning electron microscopy (SEM), larvae were exposed to *M. anisopliae* CG 153 for 24 h or 48 h. SEM confirmed the presence of fungi in the digestive tract, while HP confirmed that propagules reached the hemocoel via the midgut, damaged the peritrophic matrix, caused rupture and atrophy of the intestinal mucosa, caused cytoplasmic disorganization of the enterocytes, and degraded the brush border. Furthermore, we report for the first time the potential of *M. anisopliae* IBCB 481 to kill *Ae. aegypti* larvae and methods to improve the production of blastospores.

## 1. Introduction

*Metarhizium anisopliae* s.l. (Metsch) Sorok (Ascomycota: Clavicipitaceae) is a complex of entomopathogenic fungi (EPF) widely used as a biological pesticide to mitigate arthropod pests in agricultural settings [1]. In addition, this fungal complex has been shown to control larvae of *Aedes aegypti* [2,3,4], the principal mosquito species that transmits several arboviruses that affect millions of people around the world [5]. EPF have been used in biological control applications due to their diversity, which allows the identification of appropriate isolates to target specific insect pests [6]. Also, EPF are non-toxic to vertebrates and have limited non-target effects [7]. Thus, several researchers have proposed that EPF represent a novel approach to control insects of public health importance, such as mosquitoes [8,9,10,11,12,13,14,15,16].

Unlike other entomopathogenic organisms such as the bacteria *Bacillus thuringiensis* and the virus *Granulovirus* sp. [17] that must be ingested by the target organism, EPF can infect arthropods by penetrating directly through the tegument [18,19]. The classic infection process of *M. anisopliae* starts with the adhesion of conidia to the host cuticle that involves protein adhesions such as adhesin-like *Mad1* and *Mad2* [20]. After adhesion, the conidia produce a germ tube and deliver lipases, chitinases (*Chit1*, *Chit2*), and proteases (*Pr1*, *Pr2*) to breach and penetrate the insect cuticle [21,22]. Inside the insect hemocoel, the fungus produces yeast-like cells denominated hyphal bodies (blastospores) that deploy an extensive cocktail of extracellular metabolites such as the destruxins A, B, and C to eliminate opportunistic organisms and downregulate the innate immune system of the insect host [1,23,24]. The hyphae colonize the insect, take up nutrients from the host, and once the nutrients are exhausted, the fungi emerge from the body to produce conidia [1,20].

Although EPF classically use the tegument as the main route of infection [20] (Butt et al., 2016) they also may infect mosquitoes through the digestive or respiratory tracts [4,8,10,14,16]. In aquatic larvae, the exact mode of infection of *M. anisopliae* is still unclear. Some researchers reported that EPF conidia cannot adhere to the larval cuticle or germinate in the midgut, thus killing *Ae. aegypti* larvae by occlusion of natural orifices, the mouth, and the respiratory siphon [16,25]. Greenfield et al. [26] reported the absence of hydrocarbons in *Ae. aegypti* larvae required for the conidia to germinate. Interestingly, Alkhaibari et al. [8] reported that *M. anisopliae* ARSEF 4556 blastospores attach to the *Ae. aegypti* larval exoskeleton and enterocytes to invade the larval hemocoel for colonization. Noskov et al. [14] reported conidial development in the gut of *Aedes* larvae infected with *Metarhizium robertsii* (strain MB-1). de Oliveira Barbosa Bitencourt et al. [3] observed *M. anisopliae* ARSEF 2211 and *B. bassiana* CG 206 in the midguts of *Ae. aegypti* larvae. Thus, the route of infection seems not to be the same for all fungal strains. In fact, there is much to elucidate on how these routes vary depending on the fungal isolate, the propagules, and the insect target. 

Blastospores and conidia have distinct biological features [20]. Conidia are hydrophobic propagules produced in a solid substrate within 2 weeks and have a long shelf life [27,28]. On the other hand, blastospores are pleomorphic and hydrophilic spores that are considered analogs of hyphal bodies [20]. They can be produced in vitro [29] with shorter production cycles (~3 days) and possess morphological and physiological features to escape from insect innate immune responses [29,30]. Usually, producing a high mass of conidia is easier than producing blastospores, and the major challenge to the broad-spectrum use of blastospores as biological control agents includes production optimization and effective formulation technologies to improve their shelf life [31]. Some authors have reported the efficacy of blastospores and conidia against *Ae. aegypti* larvae [10,11,15,16]. Depending on the isolate, blastospores seem more effective against mosquito larvae than conidia [8,9,32]. However, for other isolates, blastospores, and conidia had similar virulence against *Ae. aegypti* larvae [11]. Interestingly, conidia of *Beauveria bassiana* CG 206 were slightly more virulent than blastospores against mosquito larvae. In contrast, blastospores and conidia of *M. anisopliae* CG 489 had similar results in reducing larval survival [3]. 

Despite the efficiency of EPF in invading the digestive tract of larvae, the gut is composed of a layer of epithelial cells, the enterocytes, that secrete the peritrophic matrix, digestive enzymes, and defense molecules to absorb nutrients as well as to protect themselves from pathogens [33].

Although there are reports of the presence of EPF in the digestive tract of *Ae. aegypti* larvae, it doesn’t have much information about to mechanism of action related to their ability to damage the larval gut, enter the hemocoel, and kill the larvae. We reported previously on the damage caused by EPF on the midgut of mosquito larvae [4]. Accordingly, the aims of this study were to: (1) assess the influence of shaker rotation for the production of blastospores; (2) assess and compare the virulence of blastospores and conidia of three *M. anisopliae* strains against *Ae. aegypti* larvae; (3) elucidate the damage to the larvae midgut caused by *M. anisopliae* CG 153; and (4) determine if the midgut could be an infection route for specific strains of EPF.

## 2. Materials and Methods

### 2.1. Culture and Preparation of M. anisopliae Conidia

*Metarhizium anisopliae* sensu lato (s.l.) CG 153 and CG 489 were obtained from the National Center for Genetic Resources-CENARGEN, EMBRAPA, Brazil. *Metarhizium anisopliae* s.l. IBCB 481 was obtained from the Biological Institute-IBCB, SP, Brazil. The fungal strains were cultured on potato dextrose agar medium and incubated under controlled conditions [25 ± 1 °C; relative humidity (RH) ≥ 80%]. After 15 days of incubation, the conidia were harvested and suspended in sterile 0.03% Tween 80 in dechlorinated tap water (*v*/*v*). Conidia were quantified using a Neubauer hemocytometer (KASVI^®^, São Paulo, SP, Brazil), and their concentration was adjusted to 1 × 10^8^ propagules mL^−1^ [34]. 

### 2.2. Influence of Shaker Rotations and the Time of Incubation on Blastospore Production

Three milliliters of conidial suspension at 1 × 10^8^ conidia mL^−1^ were inoculated into 125 mL Erlenmeyer flasks (N = 3) containing 42 mL of modified Adamek’s medium. The medium was prepared using 31.58 mL milhocine, 42.1 g of yeast extract (KASVI^®^, São José do Pinhais, PR, Brazil), 42.1 g of glucose D (+) anhydrous (Isofar^®^, Duque de Caxias, RJ, Brazil), and 21.03 mL of Tween 80 at 0.1% in 1000 mL of distilled water of final volume. The milhocine was prepared with 5 g of cornstarch (Maizena/Duryea©, Mogi Guaçu, SP, Brazil) dissolved in 250 mL of distilled water. The medium was prepared according to Duarte [35]. The pH was adjusted to 5.9, combining sodium hydroxide (NaOH) and hydrochloric acid (HCl) using a pH meter (mPA-201, MS Tecnopon^®^, Piracicaba, SP, Brazil]. The Erlenmeyer flasks were covered by aluminum foil and plastic wrap and then incubated at 27 °C at 150 rpm or 220 rpm (TE-424^®^, Tecnal, Brazil). After 48 h and 72 h, the medium was filtered using a sterile lint followed by centrifugation at 5000 rpm for five minutes (Rotina 380R, Hettich©, Tuttlingen, Baden-Württemberg, Germany). The supernatant was discarded, and 10 mL of 0.03% Tween 80 solution (*v*/*v*) was added. The blastospores were homogenized using a vortex (KASVI©, São José do Pinhais, PR, Brazil). The centrifugation, supernatant removal, water addition, and homogenization were conducted as described [3]. Blastospores were quantified using a Neubauer chamber [34]. The experiments were performed three different times using three different batches of propagules.

### 2.3. Insects 

The Laboratory of Physiology and Arthropods Control (Fundação Oswaldo Cruz-FIOCRUZ, Rio de Janeiro, Brazil) provided the eggs of *Ae. aegypti* (Rockefeller strain). The eggs were maintained in tap water (2 L) under controlled conditions [27 ± 1 °C; (RH) ≥ 80%] on a 12 h light/dark cycle. After hatching, the larvae were fed daily with pulverized fish food (0.05 g/L).

### 2.4. Survival Analyses

For survival analyses, blastospores and conidial suspensions were prepared and adjusted to 1 × 10^7^ propagules mL^−1^. Then, 10 *Ae. aegypti* second instar larvae (L_2_) (N = 30) were placed in plastic cups (50 mL; 4.5 cm width × 4 cm height) containing 10 mL of pure blastospores (BLA) or conidial (CON) suspensions. The control group was exposed to sterile dechlorinated tap water plus 0.03% Tween 80. The cups were covered with mesh and sealed with a rubber band and maintained under controlled conditions [27 ± 1 °C; RH ≥ 80%] on a 12:12 h light/dark cycle. The larvae were fed daily with a pinch of sterile fish food in each cup. The survival rate was monitored daily for seven days, and dead larvae were removed daily [3]. The experiments were performed three times using three batches of propagules and larvae. The tests were performed using 90 larvae per replicate (each group), totaling 630 mosquitoes for all experiments.

The fungal viability was also tested. The fungal suspensions were diluted to 1 × 10^5^ propagules mL^−1^ by serial dilution (Alves, 1998); 10 μL of conidial or blastospore suspensions were inoculated onto Petri dishes containing PDA supplemented with 0.05% chloramphenicol. PDA plates were incubated at 25 ± 1 °C and a relative humidity (RH) of ≥80%]. After 16 h, three hundred conidia or blastospores were counted randomly using an optical microscope at 400× magnification [34]. Conidia were considered to have germinated if they developed the germ tube and the blastospores if they formed hyphae [36].

### 2.5. Scanning Electron Microscopy and Histopathological Analysis

The histopathology (HP) and the scanning electron microscopy (SEM) were conducted using larvae (N = 3) exposed to *M. anisopliae* CG 153. This strain demonstrated high efficacy in killing mosquito larvae and good results in producing blastospores. Larvae were exposed to fungal propagules as described above and examined at 24 h or 48 h post-exposure by HP and SEM. To conduct HP, the larvae were fixed in Bouin’s solution [37] for 24 h at 4 °C and processed for histological examination (dehydrated in ethanol series; diaphanisation in Xylol followed by xylol-alcohol solution according to Bitencourt et al.) [3]. The blocks of paraffin were cut to 3 μm thickness, and cross-sectioned larvae were stained with hematoxylin and eosin (HE) and examined under a light microscope (Primo Star, ZEISS^®^, São Paulo, Brazil) with an integrated camera AxioCam ICc 1 (ZEISS^®^) at a magnification of 400×. 

For SEM, the larvae were fixed for 24 h in 2.5% (*v*/*v*) glutaraldehyde. The samples (N = 3) then were treated with 1% osmium tetroxide for 60 min and dehydrated through an increasing concentration series of ethanol washes as described previously [3]. The samples were photographed using a JEOL JSM-6390LV (JEOL BRASIL©, São Paulo, Brazil) scanning electron microscope at 15 kV. 

### 2.6. Statistical Analyses

Kaplan-Meier analysis was used to calculate the cumulative survival curve. Pairwise comparisons were used, and the median survival time (S_50_) was calculated using the Log-rank test. The data on the blastospore production were submitted to the Shapiro-Wilk normality test, analyzed by two-way ANOVA followed by Tukey test for multiple comparisons with a significance level of 95% (*p* ≤ 0.05). All data were analyzed by GraphPad Prism v8.00, Inc. (GraphPad Software, Boston, MA, USA). 

## 3. Results

### 3.1. Influence of Shaker Rotations and the Time of Incubation on Blastospore Production

Neither the rotation speed nor the incubation time affected blastospore production in *M. anisopliae* strains IBCB 481 and CG 153 (Figure 1). *Metarhizium anisopliae* IBCB 481 incubated at 72 h/150 rpm did not improve its blastospore production significantly compared with 48 h/150 rpm (*p* = 0.5074), 48 h/220 rpm (*p* = 0.7947), or 72 h/220 rpm (*p* = 0.2968). The blastospore production of *M. anisopliae* IBCB 481 incubated at 72 h/220 rpm did not differ significantly compared with 48 h/150 rpm (*p* = 0.9786) or 48 h/220 rpm (*p* = 0.8137). In addition, the same strain incubated at 48 h/150 rpm did not differ significantly (*p* = 0.9612) compared with 48 h/220 rpm.

The blastopore production of *M. anisopliae* CG 153 after 72 h/150 rpm was similar compared with 48 h/150 rpm (*p* = 0.9917), 48 h/220 rpm (*p* = 0.8566), and 72 h/220 rpm (*p* = 0.9989). The production of blastospores was not statistically different at 72 h/220 rpm compared with 48 h/150 rpm (*p* = 0.9726) and 48 h/220 rpm (*p* = 0.7843). *Metarhizium anisopliae* CG 153 incubated at 48 h/150 rpm did not differ significantly (*p* = 0.9563) from 48 h/220 rpm.

After 72 h of incubation, *M. anisopliae* CG 489, at 220 rpm, produced more blastospores (*p* = 0.0087) compared with 48 h/150 rpm (Figure 1), but similar numbers to 72 h/150 rpm (*p* = 0.1636) and 48 h/220 rpm (*p* = 0.2201). Here, the blastospore production was also not statistically different compared with 72 h/150 rpm vs. 48 h/150 rpm (*p* = 0.9726); 72 h/150 rpm vs. 48 h/220 rpm (*p* = 0.9988); and 48 h/150 rpm vs. 48 h/220 rpm (*p* = 4315). *Metarhizium anisopliae* CG 489 produced the most blastospores, followed by *M. anisopliae* CG 153 and *M. anisopliae* IBCB 481 (Figure 1).

### 3.2. Survival Analyzes

Only 10% of *Ae. aegypti* larvae survived after three days of exposure to *M. anisopliae* IBCB 481 blastospores. For larvae exposed to *M. anisopliae* IBCB 481 conidia, the survival rate was 21% after 3 days. At the same time, 17% and 29% of larvae survived after exposure to conidia or blastospores of *M. anisopliae* CG 153, respectively. Also, after three days, 50% and 59% of larvae survived after exposure to blastospores and conidia of *M. anisopliae* CG 489, respectively (Figure 2). By day 7, the survival rate was as follows: 0% for larvae exposed to blastospores or conidia of *M. anisopliae* CG 153 or *M. anisopliae* IBCB 481, and 40% and 47% respectively, for larvae exposed to conidia and blastospores of *M. anisopliae* CG 489 (Figure 2). Therefore, blastospores and conidia of *M. anisopliae* IBCB 481 had the lowest S_50_ of larvae, followed by *M. anisopliae* CG 153 and *M. anisopliae* CG 489. The S_50_ ranged from 1 to 7 days (Table 1).

Regardless of the fungal strain, blastospores and conidia reduced the larval survival compared with the controls (Table 1, Figure 2). However, both propagules of the *M. anisopliae* CG 489 strain were significantly less virulent against *Ae. aegypti* larvae than *M. anisopliae* IBCB 481 (*χ*^2^ = 108.5; df = 3; *p* < 0.0001) and *M. anisopliae* CG 153 (*χ*^2^ = 87.13; df = 3; *p* < 0.0001). Comparing propagules of the same strain, blastospores and conidia of *M. anisopliae* CG 489 or propagules of *M. anisopliae* CG 153 had similar results in reducing larval survival. Blastospores of *M. anisopliae* IBCB 481 were less virulent than conidia of the same strain (*χ*^2^ = 5.461; *p* = 0.0195). Blastospores of *M. anisopliae* IBCB 481 were significantly more effective as larvicides compared with either the blastospores or conidia of *M. anisopliae* CG153 (*χ*^2^ = 14.89; *p* < 0.0001 and *χ*^2^ = 5.9; *p* = 0.0151, respectively). Conidia of *M. anisopliae* IBCB 481 were not found to differ significantly from either propagule of *M. anisopliae* CG 153 (*χ*^2^ = 2.324; df = 3; *p* = 0.3129).

Considering the blastospore production and the results of the bioassays, *M. anisopliae* CG 153 was chosen for subsequent experiments.

### 3.3. Scanning Electron Microscopy and Histopathological Analysis

At both 24 h and 48 h, *M. anisopliae* CG 153 was found to have infected 100% of the larvae via ingestion (Figure 3B–F and Figure 4A–D). Propagules on the one larval cuticle, however, were observed to produce hyphae (Figure 4C,D). After 24 h and 48 h, enterocytes of all samples from the control group exhibited normal morphology, spherical nuclei located in the parabasal region, and slightly acidophilic cytoplasm. The brush border and peritrophic membranes of 100% of the larvae from the control groups were both intact (Figure 3A,B) with columnar cells. Both fungal propagules infected the insect midgut (Figure 3C–F). Larvae exposed to fungi exhibited atrophy of the intestinal mucosa (appearing as squamous enterocytes) (Figure 3C,F). Although we did not observe fungal penetration through the cuticle, there was a rupture of the intestinal mucosa and invasion of the hemocoel by blastospores in one of the larvae (33%) after 24 h of exposure (Figure 3C) in mosquitoes exposed to conidia of *M. anisopliae* CG 153. Also, we observed germ tube development by blastospores (66%-2 larvae) (Figure 3D). Forty-eight hours post-exposure to conidial suspensions, 100% of the larval gut contained several hyphal bodies, and some hyphal bodies were present in the hemocoel (Figure 3D). After 24 h, 66% of larvae exposed to blastospores of *M. anisopliae* CG 153 exhibited disrupted enterocytes without nuclei and with a disorganized and acidophilic cytoplasm. Also, the brush border showed significant damage with an irregular surface or a lack of brush on the enterocytes (100% of the larvae-3 larvae) (Figure 3E). By 48 h, the three larvae exposed to blastospores showed flattened enterocytes (Figure 3D). The peritrophic matrix and extracellular material normally produced by enterocytes were not observed at either 24 h or 48 h in all larvae exposed to fungal propagules.

## 4. Discussion

Biocontrol strategies involving *M. anisopliae* represent a new weapon against larvae of the principal dengue vector, *Ae. aegypti* [8,9,13] (Alkhaibari et al., 2016, 2017; De Paula et al., 2021). We demonstrate that *M. anisopliae* CG 153 caused significant damage to the midgut integrity, affecting the larval enterocytes, which may explain the virulence of this fungus on larvae through a non-classical route. Furthermore, our experiments demonstrated the substantial effectiveness of three strains of *M. anisopliae* (CG 153, CG 489 and IBCB 481) against *Ae. aegypti* larvae.

We also modified the production parameters and improved the production of blastospores in one fungal strain (*M. anisopliae* CG 489). Blastospores are yeast-like cells produced in vitro conditions in a short incubation time [31]. Here, blastospore production increased after extending the incubation time [38]. However, only *M. anisopliae* CG 489 improved blastospore production at 200 rpm. This result suggests that although the three strains belong to the genus *Metarhizium*, the strains have intra-specific variations with specific features and requirements for energy resources that distinguish them from each other, which might directly affect their blastospore production [39,40]. Therefore, additional studies should be conducted to understand these intrinsic fungal characteristics and requirements to improve blastospore production.

Blastospores of *M. anisopliae* CG 489 demonstrated slightly better efficacy than conidia in reducing the survival of *Ae. aegypti* larvae, corroborating the results of Alkhaibari et al. [8] (2016). However, blastospores and conidia of *M. anisopliae* CG 153 and *M. anisopliae* IBCB 481 had similar effectiveness, corroborating the reports of de Oliveira Barbosa Bitencourt et al. [3,11] (2018, 2021a). Blastospores are in vitro propagules analogous to hyphal bodies and rapidly colonize aquatic targets such as mosquito larvae [8,41]. Although conidia are known to colonize terrestrial invertebrates [1], our findings demonstrate that they also can infect mosquito larvae in the aquatic environment and reduce larval survival. However, the efficacy of blastospores or conidia also depends on the genus and strain of the fungus, environmental conditions, and the target species [20].

Both propagules of *M. anisopliae* CG 153 use the mosquito’s digestive tract as the main front of interaction, as described previously [3,42]. However, this fungus also colonizes the hemocoel [8,14]. In addition, fungal propagules damaged the enterocytes and brush borders, which also might lead to the death of the larvae and invasion of the hemocoel. Interestingly, Silva et al. [43] reported epithelial tissue degeneration in the larval gut of *Aedes albopictus* exposed to *Bacillus thuringiensis*, and Jiraungkoorskul [44] and Al-Mehmadi and Al-Khalaf [45] reported similar results using plant extracts against *Culex quinquefasciatus*.

The histological analyses revealed a high concentration of hyphae and propagules inside the midgut, which caused damage to the intestinal mucosa, peritrophic membrane, necrosis of enterocytes, and intestinal disruption. The *Ae. aegypti* midgut is composed of absorptive enterocytes and non-absorptive hormone-producing cells [46]. Enterocytes play an important role in insect physiology, contributing to digestion, homeostasis, and protection against invading microorganisms, such as the formation of a peritrophic matrix, production of reactive oxygen species, as well as cell proliferation and differentiation [47,48,49]. Additionally, the digestive tract is a niche for microbiota. Taracena et al. [48] reported the damage to enterocytes and, at the same time, the proliferative ability of the intestinal cells of *Ae. aegypti* adults after being infected with the Dengue virus. In addition, Buchon et al. [50,51] reported that the presence of the microbiota and pathogenic bacteria increases the rate of epithelial turnover. However, substantial destruction of the enterocytes could result in a fatal loss of gut barrier integrity [40]. Taracena et al. [48] also observed an increase in mitosis of the peritrophic matrix after mosquito adults were exposed to the entomopathogenic bacteria *Pseudomonas entomophila*. Enterocytes produce a type 1 peritrophic matrix for adults and type 2 in larvae [52,53]. Interestingly, we did not observe a peritrophic matrix in larvae exposed to *M. anisopliae* CG 153. Our data suggest that the midgut serves as a gateway for fungi to colonize the hemocoel. Once the mucosa is damaged, the digestive process of the larvae becomes affected by interaction with this fungus, causing death.

Besides damaging the enterocytes, EPF might produce several secondary metabolites, such as the toxins destruxins and beauvericin [1,4,54]. Destruxins have insecticidal activity and play a crucial role in blocking calcium channels causing flaccid paralysis and inhibiting host immune responses [23,55]. Furthermore, Dumas et al. [56] reported the cytotoxic effect of destruxin E on brush border and epithelial cells of the *Galleria mellonella* midgut. Although we did not conduct experiments on the production of toxins, we suggest that *M. anisopliae* CG 153 might deliver secondary metabolites into the larval midgut, delaying enterocyte proliferation and differentiation. Additionally, the toxins would contribute to the degeneration of the brush borders, reduced secretion of the peritrophic matrix, and affect the gut microbiota, which also affects the enterocytes. Together these effects of the EPF on midgut physiology could explain the pathogenesis we observed.

## 5. Conclusions

Control of mosquitoes is essential to reduce the transmission of vector-borne diseases. Therefore, new biocontrol approaches to control *Ae. aegypti* adults and larvae need to be developed, and EPF have demonstrated their potential as effective insecticides. Here, low speed and shorter incubations only reduced the blastospores yield of *M. anisopliae* CG 489, providing new information on how blastospores might be produced on a larger scale. Blastospores and conidia of *M. anisopliae* CG 489, *M. anisopliae* CG 153, and *M. anisopliae* IBCB 481 showed strong potential as larvicidal agents. Additionally, we demonstrated the digestive tract as the major route of infection in the aquatic environment. Furthermore, we have shown details of enterocyte damage and brush border degradations in mosquito larvae exposed to EPF, which might explain the virulence of *M. anisopliae* CG 153. Although this isolate cannot successfully infect the mosquito’s integument, it can enter through the digestive tract, and move to the hemocoel and kill the larvae, reinforcing its potential use against *Ae. aegypti* larvae.

## Figures and Tables

**Figure 1 insects-14-00328-f001:**
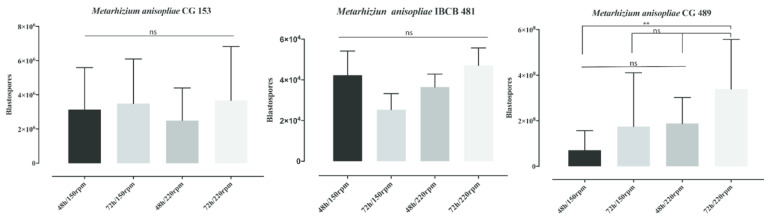
The production of blastospores of three different *Metarhizium anisopliae* s.l strains under different shaker rotations and times. Significance levels were calculated, and their strength was displayed with the symbol (** *p* < 0.001; ns = not significant).

**Figure 2 insects-14-00328-f002:**
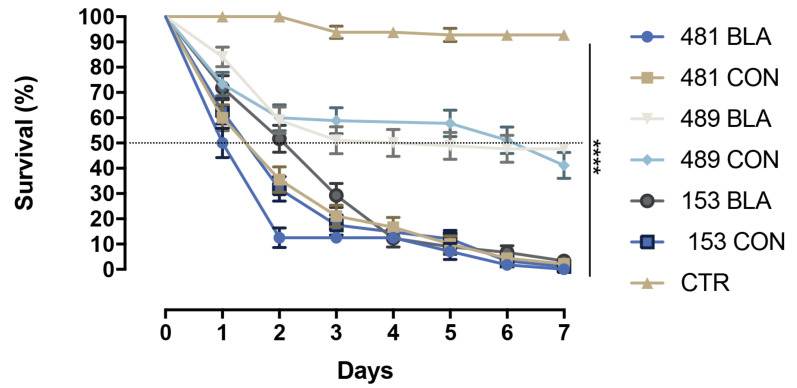
Survival curves of *Aedes aegypti* larvae exposed to conidia (CON) or blastospores (BLA) of *Metarhizium anisopliae* s.l strains or exposed to Tween 80 at 0.03% (CTR). Significance levels were calculated, and their strength was displayed with the symbol (**** *p* < 0.0001) compared with the control group.

**Figure 3 insects-14-00328-f003:**
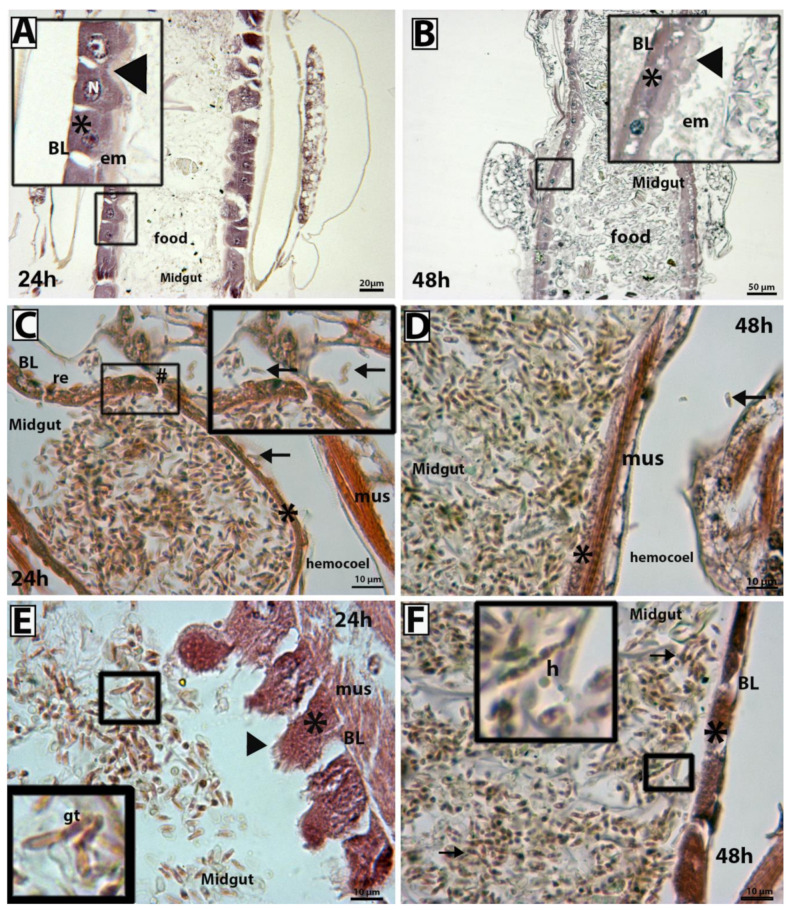
Histological analyses of *Aedes aegypti* larvae, stained with Hematoxylin and Eosin. Photomicrographs of *Ae. aegypti* larva midgut at 24 h or 48 h after exposure to propagules of *Metarhizium anisopliae* s.l CG 153 or Tween 80 at 0.03% for 24 h (CTR). Tween 80 at 0.03% for 24 h (**A**) and 48 h (**B**); conidia at 24 h (**C**) and 48 h (**D**); and blastospores at 24 h (**E**) and 48 h (**F**). Larvae from the control group exhibited normal midguts composed of healthy enterocytes with acidophilic cytoplasm, preserved brush border and normal extracellular material. After 24 h or 48 h, larvae exposed to conidia presented disrupted midguts with flattered enterocytes without brush borders and had hyphal bodies in the hemocoel. After 24 h, larvae exposed to blastospores showed enterocytes with disorganized cytoplasm and disrupted brush borders (black arrowhead). After 48 h, larvae exposed to blastopores exhibited midguts with flattered enterocytes without brush borders and the presence of hyphae in the gut lumen. BL = basal lamina; N = nucleus; * = cytoplasm of the enterocyte; brush border (black arrowhead); hyphae (h); enterocytes (asterisk); germ tube (gt); muscle (mus), disrupted midgut (#).

**Figure 4 insects-14-00328-f004:**
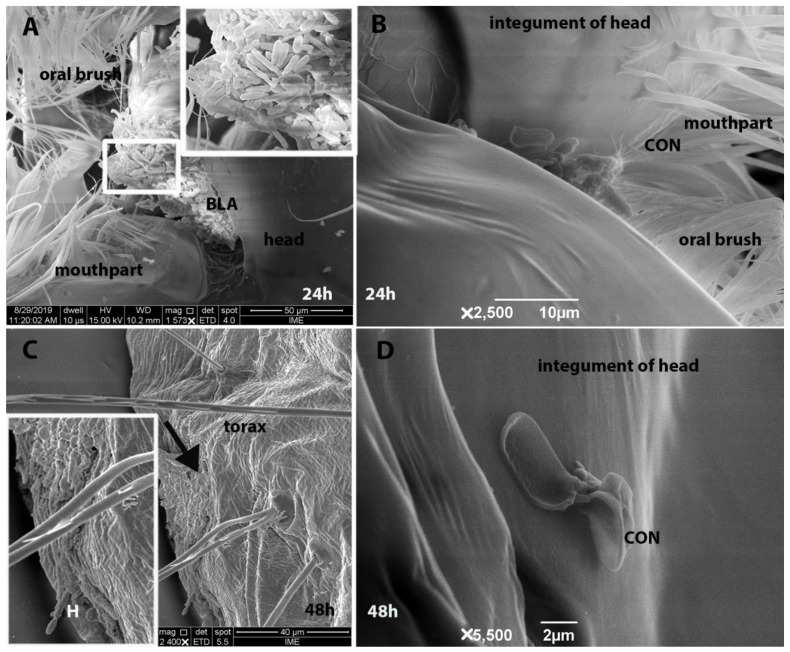
Scanning electron microscopy of *Aedes aegypti* larvae exposed for 24 h and 48 h to conidia or blastospores of *Metarhizium anisopliae* s.l CG 153 strain. (**A**) Blastospores at 24 h (BLA); (**B**) blastospores at 48 h (BLA); (**C**) conidia at 24 h (CON); and (**D**) conidia at 48 h (CON); hyphae (H).

**Table 1 insects-14-00328-t001:** Log-rank pairwise comparisons of survival rates and median survival time (S_50_) of *Aedes aegypti* 2nd instar larvae exposed to *Metarhizium anisopliae* strains or exposed to Tween 80 at 0.03%.

Groups	1 × 10^7^ Propagules mL^−1^							S_50_
	CTR	481 BLA	481 CON	489 BLA	489 CON	153 BLA	153 CON	
CTR	-	*χ*^2^ = 192.3; *p <* 0.0001	*χ*^2^ = 243.3; *p* < 0.0001	*χ*^2^ = 48.10; *p* < 0.0001	*χ*^2^ = 61.35; *p* < 0.0001	*χ*^2^ = 178.1; *p* < 0.0001	*χ*^2^ = 197.0; *p* < 0.0001	ND
481 BLA	-	-	*χ*^2^ = 5.461; *p* = 0.0195	*χ*^2^ = 61.27; *p* < 0.0001	*χ*^2^ = 53.26; *p* < 0.0001	*χ*^2^ = 14.89; *p =* 0.0001	*χ*^2^ = 5.900; *p =* 0.0151	1
481 CON	-	-	-	*χ*^2^ = 47.50; *p* < 0.0001	*χ*^2^ = 46.53; *p* < 0.0001	*χ*^2^ = 46.53; *p* < 0.0001	*χ*^2^ = 1.574; *p =* 0.2096	2
489 BLA	-	-	-	-	*χ*^2^ = 0.887; *p =* 0.3463	*χ*^2^ = 34.24; *p* < 0.0001	*χ*^2^ = 50.33; *p* < 0.0001	5.5
489 CON	-	-	-	-	-	*χ*^2^ = 35.43; *p* < 0.0001	*χ*^2^ = 48.25; *p* < 0.0001	7
153 BLA	-	-	-	-	-	-	*χ*^2^ = 2.264; *p =* 0.1324	3
153 CON	-	-	-	-	-	-	-	2

Statistical significance (*p* value) between *Aedes aegypti* larvae exposed to *Metarhizium anisopliae* CG 489, *Metarhizium anisopliae* CG 153 or *Metarhizium anisopliae* IBCB 481 strains. BLA = blastospores; CON = conidia; CTR = control group. ND = not determined and *χ*^2^ = Chi-square value.

## Data Availability

All the data generated from the current work are available upon reasonable request to the corresponding authors.

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
