# Peer review of "A Novel Model of Pathogenesis of Metarhizium anisopliae Propagules through the Midguts of Aedes aegypti Larvae"

_insects, 2023, doi:10.3390/insects14040328_

Round 1
Reviewer 2 Report
The manuscript shows very significant results regarding the pathology description of Metarhizum anisopliae when infecting larvae of the mosquito Aedes aegypti. In fact, through the paper, many findings are documented. The data generated make it worthy of publishing it. However, respecting the author's style, I may have suggestions to follow the order of ideas better.
1. First, a logical sequence through the sections for readers is not detected.
2. Increasing the aims numbers would facilitate describing the bulk results. For instance, a 4th or even a 5th aim would greatly help to order the whole manuscript; i.e., aim 1) influence of shaker rotation for the production conditions?, and to separate a 5th aim as 5) description of HP and SEM fungi damage in the larval body after ingestion? (without deleting the already proposed aims).
Line 111: strains under conditions... which conditions?
Line 112: and finally.. and finally, 3)? Do the authors want to add here a third aim?
My second significant suggestion deals with increasing quantitative versus qualitative descriptions of some significant results. Section 3.3 is a good example. Authors only describe specific organs and tissue damages associated with the fungi after ingestion. It includes the group of pictures. How many larvae are presented in each particular histopathology? Just one? 100% of them? And so on. Simple percents would restrengthen the valuable author's scientific findings.
